# Sex as Biological Variable in Cardiac Mitochondrial Bioenergetic Responses to Acute Stress

**DOI:** 10.3390/ijms23169312

**Published:** 2022-08-18

**Authors:** Susan R. Scott, Kanhaiya Singh, Qing Yu, Chandan K. Sen, Meijing Wang

**Affiliations:** 1Department of Surgery, Indiana University School of Medicine, Indianapolis, IN 46202, USA; 2Indiana Center for Regenerative Medicine and Engineering, Indiana University School of Medicine, Indianapolis, IN 46202, USA

**Keywords:** mitochondrial bioenergetics, sex-specific differences, acute inflammation, oxidative stress

## Abstract

Cardiac dysfunction/damage following trauma, shock, sepsis, and ischemia impacts clinical outcomes. Acute inflammation and oxidative stress triggered by these injuries impair mitochondria, which are critical to maintaining cardiac function. Despite sex dimorphisms in consequences of these injuries, it is unclear whether mitochondrial bioenergetic responses to inflammation/oxidative stress are sex-dependent. We hypothesized that sex disparity in mitochondrial bioenergetics following TNFα or H_2_O_2_ exposure is responsible for reported sex differences in cardiac damage/dysfunction. Methods and Results: Cardiomyocytes isolated from age-matched adult male and female mice were subjected to 1 h TNFα or H_2_O_2_ challenge, followed by detection of mitochondrial respiration capacity using the Seahorse XF96 Cell Mito Stress Test. Mitochondrial membrane potential (ΔΨm) was analyzed using JC-1 in TNFα-challenged cardiomyocytes. We found that cardiomyocytes isolated from female mice displayed a better mitochondrial bioenergetic response to TNFα or H_2_O_2_ than those isolated from male mice did. TNFα decreased ΔΨm in cardiomyocytes isolated from males but not from females. 17β-estradiol (E2) treatment improved mitochondrial metabolic function in cardiomyocytes from male mice subjected to TNFα or H_2_O_2_ treatment. Conclusions: Cardiomyocyte mitochondria from female mice were more resistant to acute stress than those from males. The female sex hormone E2 treatment protected cardiac mitochondria against acute inflammatory and oxidative stress.

## 1. Introduction

Cardiac damage and dysfunction following trauma, sepsis, and heart surgery (having myocardial ischemia) largely impact clinical outcomes [1,2,3,4,5,6,7]. Over 50% of all critically injured trauma patients requiring intensive care unit (ICU) treatment develop cardiovascular dysfunction, which contributes to 20% mortality [5,8]. Septic patients with cardiovascular dysfunction are three to four times more likely to die than those without cardiac damage [6,7]. Moreover, about 40% of the incidence of new irreversible myocardial injury occurs in patients who underwent heart surgery with cardiopulmonary bypass [9]. Such cardiac damage is associated with increased mortality, slowed recovery, and longer ICU stay and hospitalization [1,2,3,4]. To date, the precise cause of cardiac damage and dysfunction remains elusive during these acute injuries. Therefore, gaining a better understanding of the pathophysiological process underlying cardiac dysfunction and damage is critical for developing effective therapeutic approaches to improve patient prognosis.

Given that the heart is a high-energy-demanding organ, energy metabolism plays a central role in heart pathophysiology [10]. Mitochondria, one of the most important subcellular organelles in cardiomyocytes, provide 90% of the energy required for maintaining normal cardiac function and are central to heart bioenergetics. Therefore, mitochondria are a key determinant of cardiac pathophysiology. Inflammatory mediators, such as tumor necrosis factor α (TNFα), and reactive oxygen species (ROS) have been shown to impair mitochondrial function [11,12]. TNFα and ROS significantly increase preceding cardiac damage and dysfunction following trauma, sepsis, and myocardial ischemia [13,14,15,16,17,18,19,20]. Local cardiac inflammatory response and oxidative stress have been demonstrated as the primary driving force for cardiomyocyte impairment and functional damage following these injuries [8,19,21,22,23,24,25,26]. On the other hand, sex dimorphism impacts consequences after trauma, sepsis, or myocardial ischemia [27,28,29,30,31,32]. Women, specifically young females, demonstrate a more favorable outcome and decreased mortality following traumatic injuries [31,33,34,35,36,37]. Epidemiological studies also consistently report that the male sex is an important risk factor for severe sepsis with developed organ dysfunction [38,39,40,41]. Our previous studies have further shown sex disparities in myocardial responses to global ischemia with better recovery in female hearts compared to male ones [25,42,43,44,45]. Of note, the influence of sex has been observed on the expression of genes related to cardiac energy metabolism and mitochondria [46]. Collectively, we propose that sex-specific mitochondrial metabolic response to inflammatory mediators or ROS may be one of the underlying mechanisms for sex disparities in cardiac damage/dysfunction.

We have shown that TNFα depresses cardiac function and that females are more tolerant to TNFα-induced cardiac dysfunction than males [42,47,48]. Sex differences exist in mitochondrial membrane potential and ROS generation in adult cardiomyocytes exposed to hydrogen peroxide (H_2_O_2_, the most stable form of ROS) [45]. However, it is unclear whether sex as a biological variable influences mitochondrial bioenergetic profile of cardiomyocytes in response to TNFα or H_2_O_2_. In this study, we determined mitochondrial metabolic function in cardiomyocytes isolated from male and female mice upon exposure of TNFα or H_2_O_2_ using the extracellular flux (XF) methodology (Seahorse Cell Mito Stress test) and evaluated the role of the female hormone estrogen in regulating the cardiomyocyte mitochondrial bioenergetic profile during acute stress.

## 2. Results

### 2.1. Effects of Cell Density and Carbonyl Cyanide-4 (Trifluoromethoxy) Phenylhydrazone (FCCP) Concentration on Mitochondrial Bioenergetic Response

To acquire reliable oxygen consumption rate (OCR) measurements, we first determined the optimal cardiomyocyte seeding density. The linear OCR values were observed between cell densities of 6000 and 20,000 cells per well seeded overnight on a 24-well Seahorse plate [49]. In addition, 1000 cells per well using a 24-well Agilent Seahorse plate were suitable to measure mitochondrial metabolic rate in cardiomyocytes a couple of hours after their isolation [50]. In this study, we performed the XF Mito Stress test several (three to six) hours after cardiomyocyte isolation using a 96-well Agilent Seahorse V3 plate. Therefore, we tested cell density of 1500 or 3000 cells per well. We did not see a difference in basal (State III respiration) OCR values between these two groups in cardiomyocytes isolated from adult male and female mice (Figure 1A). Based on the Agilent Seahorse XF recommendation, basal OCR ranges would be targeted between 20 and 165 pmol/min to avoid potential floor or ceiling values after using mitochondrial respiration modulators. Additionally, to minimize the effect of losing unattached cells on mitochondrial bioenergetic measurements, we selected higher cardiomyocyte density (3000 cells/well) that fell into the recommended basal OCR range in the rest of this study.

To determine the optimal dose of FCCP, we conducted an FCCP titration experiment. We used 0, 0.5, 0.75, 1, and 1.5 µM of FCCP in different cell density assays. Cardiomyocytes without the FCCP treatment failed to display a maximal bioenergetic response (Figure 1B). We found that 0.75 µM of FCCP was suitable to induce a maximal respiration response in both 1500 and 3000 cells/well (Figure 1C,D) and used this FCCP dose for the rest of our experiments. The OCR trace with different doses of FCCP is shown in Appendix A. Cardiomyocytes were detected instantly prior to the assay (Appendix A).

### 2.2. Mitochondrial Bioenergetic Function between Cardiomyocytes Isolated from Adult Male and Female Mice

There were no differences in functional measurements of respiratory chain activities in mitochondria isolated from male and female rat hearts [51]. To determine the role of sex in mitochondrial bioenergetic function in a cellular context, we performed the Seahorse Cell Mito Stress test in intact cardiomyocytes isolated from age-matched adult male and female mice in this study. Based on six individual Seahorse assays on cardiac myocyte isolated from six male and six female mouse hearts, we observed comparable OCR values of basal, non-mitochondrial, and maximal respiration capacities in adult cardiomyocytes from male and female mice (Figure 2A,B).

### 2.3. Sex Differences in Cardiac Mitochondrial Bioenergetic Response to TNFα or H_2_O_2_

Our published study has revealed that there are sex disparities in TNFα-induced cardiac dysfunction [42,47,48] and in oxidative stress (H_2_O_2_)-damaged mitochondrial membrane potential (ΔΨm) in cardiomyocytes [45]. We thus determined mitochondrial metabolic changes following TNFα or H_2_O_2_ exposure in cardiomyocytes isolated from age-matched adult male and female mice. According to our previous work in which serum TNFα levels were ~11.5 ng/mL at 1 h after a lipopolysaccharide challenge in vivo [52] and a TNFα dose-response study (Figure 3A), we selected 10 ng/mL of TNFα in our experiments. Decreased OCR values of basal and non-mitochondrial respiration but increased maximal respiratory rate and spare capacity were observed in cardiomyocytes isolated from female mice when compared to those from males upon 10 ng/mL of TNFα stimulation (Figure 3B,C). The choice of H_2_O_2_ used at 50 μM for 1 h was based on our previous work, showing that this dose of H_2_O_2_ significantly reduced mitochondrial ΔΨm but caused negligible cell death in cardiomyocytes isolated from male adult mice [45]. Notably, cardiomyocytes isolated from female mouse hearts displayed higher levels of maximal mitochondrial respiratory function in response to H_2_O_2_ compared to those of male cardiomyocytes (Figure 4A,B).

### 2.4. Effect of 17β-Estradiol (E2) on Mitochondrial Bioenergetic Function in Adult Cardiomyocytes Isolated from Male Mice

Among the factors contributing to sex differences, estrogen plays a key role in mediating sex disparities in the cardiovascular system [53,54]. Therefore, we assessed the effect of E2 on mitochondrial respiration performance in cardiomyocytes isolated from male adult mice following TNFα or H_2_O_2_ exposure. The dose of E2 was selected according to our previous studies [45,55,56]. We found that TNFα significantly increased the non-mitochondrial respiration rate and reduced maximal respiration capacity, while E2 treatment restored the TNFα-impaired mitochondrial bioenergetic response in cardiomyocytes from male mice (Figure 5A). Decreased maximal OCR value was observed in H_2_O_2_-stimulated cardiomyocytes. Intriguingly, H_2_O_2_-disrupted mitochondrial metabolic function was also protected by E2 usage (Figure 5B). E2 did not affect the mitochondrial bioenergetic response in cardiomyocytes isolated from male adult mice without stress (Appendix A). 

### 2.5. Alteration of Mitochondrial Membrane Potential in TNFα-Treated Cardiomyocytes

The mitochondrial ΔΨm generated by proton pumps (Complexes I, III, and IV) plays a key role in mitochondrial homeostasis and is essential for maintaining mitochondrial metabolic function. We have shown that H_2_O_2_ damaged mitochondrial ΔΨm, which was preserved by E2 treatment in cardiomyocytes isolated from male mice [45]. Given sex differences in the TNFα-disrupted mitochondrial bioenergetic response, we further determined the effect of TNFα on mitochondrial ΔΨm in cardiomyocytes isolated from male and female mice. We observed that TNFα markedly decreased mitochondrial ΔΨm in cardiomyocytes from male mice but not from females (Figure 6B). Importantly, improved mitochondrial ΔΨm was noticed in E2-treated cardiomyocytes from male mice upon TNFα exposure (Figure 6C), while E2 treatment did not impact mitochondrial ΔΨm in female cardiomyocytes with TNFα stimulation (Appendix A). Representative images of JC-1-stained cardiomyocytes from each experimental group are shown in Appendix A.

## 3. Discussion

This is the first study to determine sex differences of bioenergetic profiling in isolated intact adult mouse cardiomyocytes subjected to TNFα or H_2_O_2_ exposure using a high-throughput XF analyzer. Here, our results indicated that cardiomyocytes from female mice displayed a better mitochondrial bioenergetic response to TNFα or H_2_O_2_ with higher maximal respiratory capacity than those from male mice. In addition, E2 treatment improved mitochondrial metabolic function in cardiomyocytes isolated from male mice following TNFα or H_2_O_2_ treatment. These data suggest that the benefit in cardiomyocytes from female animals may be attributable to estrogen-derived mitochondrial protection, at least in part.

Mitochondria mainly function as the high-energy molecule ATP factory in eukaryotic cells and are abundant in cardiomyocytes, comprising about 35% of myocyte volume in the heart [57]. The normal heart function relies on tremendous energy from oxidative phosphorylation that is largely met via mitochondrial metabolism. Herein, we employed the Seahorse XF extracellular flux technology to analyze mitochondrial respiration function in adult cardiomyocytes. In addition to gaining the basal respiration rate, Seahorse XF analysis provides important information on key parameters of mitochondrial bioenergetic profiling using respiration modulators as follows: (1) oligomycin to inhibit ATP synthase (complex V) for acquiring the fraction of oxygen consumption linked to cellular ATP production; (2) FCCP, an uncoupling agent, to disrupt mitochondrial ΔΨm through breaking down the proton gradient, thus resulting the maximal oxygen consumption by complex IV; and (3) combination of rotenone (complex I inhibitor) and antimycin A (complex III inhibitor) to shut down mitochondrial respiration, enabling the estimation of non-mitochondrial respiration.

The use of oligomycin and the dose of the uncoupler, FCCP, are important to determine respiratory capacity. Oligomycin blocks the proton channel of ATP synthase, inhibiting protons back into the mitochondria and thus decreasing electron flow to reduce mitochondrial respiration. The presence of oligomycin also prevents the reverse activity of ATP synthase due to rapid depletion of intracellular ATP during calculation of maximal respiration (after FCCP addition). While the use of oligomycin following basal measurements decreases mitochondrial respiration in most cells, we found that it did not impact OCR in adult cardiomyocytes in the present study. This finding is in line with a previous observation that mitochondrial oxygen consumption and the substrate-consuming process during basal respiration are mainly mediated by proton leak in isolated cardiomyocytes from other groups [49,50] and likely due to reduced ATP demand from inhibition of myocyte contraction during basal respiration in quiescent cardiomyocytes. Of note, a higher dose of oligomycin has been reported to disturb the subsequent response to FCCP, thus decreasing maximal OCR [49,58]. The choice of 1 µM of oligomycin used in this study was based on previous studies showing the suitability of this dose in adult cardiomyocytes [49]. Therefore, here it was sufficient to block ATP synthase but not to reduce the maximal respiration rate. On the other hand, an uncoupler of FCCP is used to estimate maximal capacity of the mitochondrial electron transport chain (ETC). After mitochondrial ΔΨm is collapsed by FCCP, electron flow is uninhibited through the ETC, and oxygen consumption reaches the maximum. Although FCCP is often used at 1 µM in the Mito Stress Test assay [49], it displays a bell-shaped dose-response curve due to excess FCCP (over the optimal uncoupling concentration) inhibiting respiration [59]. Thus, it is necessary to determine the optimal dose of FCCP in different cell populations. Herein, we found 0.75 µM of FCCP able to support maximal uncoupled respiration in adult cardiomyocytes. A decrease in FCCP-stimulated respiration is a strong indicator of potential mitochondrial malfunction [60].

While sex differences have been observed in mitochondrial ROS production/oxidative damage [61,62,63] and in calcium-induced mitochondrial swelling (mitochondrial permeability transition pore opening) [51,64], there is little information available regarding the role of sex in mitochondrial bioenergetic function in adult cardiomyocytes. It is noteworthy that sex-specific differences have been reported in metabolic responses following acute injury, such as trauma, shock, and sepsis [65,66]. In the present study, we demonstrated the initial evidence showing the impact of sex as a biological variable on cardiac mitochondrial metabolic alterations upon TNFα or H_2_O_2_ stress. We observed that during normal conditions, there were no differences in the mitochondrial bioenergetic response in cardiomyocytes isolated from age-matched adult male and female mice. In fact, comparable mitochondrial respiratory chain activities in substrate oxidation and coupled ATP generation have been reported in mitochondria from male and female rat hearts, with similar amounts of respiratory chain complexes I–IV [51]. However, there was better mitochondrial metabolic function with higher maximal respiration capacity in cardiomyocytes from female mice when compared to those from male ones in response to TNFα or H_2_O_2_, indicating better performance and less damage in cardiomyocytes from female animals during such stress. These findings provide the mechanistic evidence in support of the female heart resistant to TNFα-depressed cardiac function and to cardiac dysfunction induced by trauma/sepsis/ischemia (increased inflammatory cytokines and ROS).

Despite sex differences attributable to multiple aspects, the sex hormone estrogen is a key factor in mediating the female myocardial response to injury [53,54]. Myocardial ischemia occurs uncommonly in premenopausal women, whereas this risk increases in the postmenopausal age group [67,68], suggesting that the presence of estrogen likely protects the heart from ischemic injury. Studies from our group and others have shown that exogenous estrogen supplementation provides cardioprotection in male and ovariectomized female animals following acute myocardial ischemia/reperfusion (I/R) injury or trauma hemorrhage [44,69]. With respect to the influence of E2 on mitochondrial function, investigations mainly focus on the transcriptional regulation of E2 in expression of nuclear-encoded mitochondrial respiratory complex genes and mitochondrial DNA through estrogen receptors [70]. In fact, E2 can directly localize to the mitochondrial membrane, decreasing microviscosity and improving bioenergetic function in skeletal muscle [71]. Estrogen has also been reported to interact with ATP synthase, thus regulating cellular energy metabolism [72]. We have further observed that acute E2 treatment protected mitochondrial ΔΨm in cardiomyocytes from male adult mice and reduced mitochondrial ROS production following I/R or an H_2_O_2_ challenge [45]. In this study, we extended our findings to the effect of rapid E2 usage on the mitochondrial bioenergetic response to acute stress (TNFα or H_2_O_2_ stimulation here). We found that concomitantly using E2 (1 h) significantly restored TNFα- or H_2_O_2_-impaired mitochondrial maximal respiratory function in cardiomyocytes isolated from adult male mice. Improved mitochondrial spare capacity was also noticed in E2-treated cardiomyocytes. Notably, during stress, energy demand increases, and more ATP is needed to maintain cellular functions. Larger spare respiratory capacity implies that more ATP can be produced in a cell to overcome more stress. Therefore, our results suggest that E2 treatment improves the cardiomyocytes’ ability to meet largely increased ATP turnover upon acute stress. It is noteworthy that the rapid use of estrogen post-injury provided therapeutic potential following trauma–hemorrhage [73,74,75], burn [76,77], and sepsis [78,79]. Our group also demonstrated the beneficial effect of acute post-ischemic E2 treatment on heart I/R [45,80]. More importantly, one-dose usage of E2 right after a burn injury significantly improved cardiac function and preserved mitochondrial performance within 24 h [77]. We further observed that E2 treatment post ischemia corrected male cardiomyocyte mitochondrial activity following I/R [45]. Together with our current findings, these studies strongly support considering the rapid use of E2 as an adjunct therapy for patients under acute stress.

The values of key parameters in mitochondrial bioenergetic profiling relies on a normal proton gradient, which is formed by mitochondrial ΔΨm. Disrupted mitochondrial ΔΨm by TNFα or H_2_O_2_ collapses the proton gradient prior to FCCP addition, thus leading to reduced maximal uncoupled respiration. Our previous study has indicated that H_2_O_2_ impairs mitochondrial ΔΨm in cardiomyocytes from male mice [45], along with a decreased maximal OCR value here. We further observed that TNFα significantly reduced mitochondrial ΔΨm in cardiomyocytes from male mice (not from female animals), associated with lower maximal respiration capacity in the present study. E2 treatment restored TNFα-damaged mitochondrial ΔΨm and maximal respiratory response in cardiomyocytes from male mice. These data confirmed the importance of mitochondrial ΔΨm in maintaining mitochondrial bioenergetic function.

It is evident that antibiotics (streptomycin and gentamicin) are toxic to mitochondria. Cells, including prostate cancer cells, human lymphoblastoid cells, and hepatocytes, cultured in the presence of streptomycin do not maintain oxidative metabolism, and mitochondria isolated from streptomycin-treated cells do not display respiration on any substrate [81]. In this regard, we performed experiments to determine the effect of Pen/Strep on mitochondrial respiration in isolated cardiomyocytes. Our results indicated that comparable respiration function was observed in cardiomyocytes cultured in the presence or absence of Pen/Strep (Appendix A), suggesting that Pen/Strep, at least in short-time use, do not affect the mitochondrial bioenergetic response in intact cardiomyocytes.

A supraphysiological concentration of pyruvate (1 mM) was present in the assay medium in this study [49], leading to a bioenergetic response of adult cardiomyocytes independent of other substrates (i.e., fatty acid, glucose). Despite 60–90% of energy demands in the healthy heart for oxidative phosphorylation from fatty acid oxidation, a metabolic shift with greater rate of lipid oxidation and decreased glucose oxidation has been observed in heart-failure patients [10,82]. Considering sex-related differences in fatty acids’ metabolism [46,83], it would be of great interest to examine the role of sex on substrate-dependent mitochondrial respiratory function during acute stress in the future. To this end, pyruvate titration is required to be performed at more physiological levels [49]. In the current study, the XF Cell Mito Stress Test was used to assess mitochondrial bioenergetic function. If the work focuses on glycolytic activity in adult cardiomyocytes during stress, the acidification rate (ECAR) needs to be analyzed using the XF Glycolysis Rate Assay. Similarly, ATP production from mitochondrial respiration and glycolysis can also be measured in isolated cardiomyocytes under pathophysiological conditions using an Agilent Seahorse XF Real-Time ATP Rate Assay kit in the future. Of note, individual ETC complexes can be assembled into supercomplexes (SCs) to transfer electrons/substrates more efficiently, to protect individual ETC complexes, to reduce ROS production, and to coordinate alterations in cellular metabolism [84]. Emerging evidence has suggested that deficiency/disruption of SCs is associated with myocardial ischemia and heart failure [85,86]. In the present study, we did not evaluate the role of sex in modulating the formation of mitochondrial SCs in cardiomyocytes during acute stress. However, particular interest will be given to this important unknown in our future investigation.

## 4. Materials and Methods

### 4.1. Animals

Male and female C57BL/6J mice were purchased from Jackson Laboratories (Bar Harbor, ME, USA). All mice were acclimated for at least 5 days with a standard diet before the experiments. A total of 24 mice (14 males and 10 females) at 11–22 weeks of age were used for the experiments.

### 4.2. Adult Mouse Cardiomyocyte Isolation and Preparation

A Langendorff perfusion system was used to isolate single cardiomyocytes from adult male and female mouse hearts as we described previously [45,87]. Briefly, after being injected with heparin (100 IU, i.p.), the mice were euthanized with an overdose of isoflurane. The hearts were excised rapidly and transferred to a Langendroff unit immediately. The hearts were retrogradely perfused with a calcium-free perfusion buffer (NaCl 113 mM, NaH_2_PO_4_ 0.6 mM, NaHCO_3_ 1.6 mM, KCl 4.7 mM, KH_2_PO_4_ 0.6 mM, MgSO_4_ 1.2 mM, HEPES 10 mM, Taurine 30 mM, 2,3-butanedione monoximoe [BDM] 10 mM, and glucose 20 mM, pH 7.4) for 2–3 min. The hearts were then digested with collagenase II (1.5 mg/mL) for 11–13 min. Isolated cardiomyocytes were sequentially restored in a perfusion buffer containing calcium (100, 250, 500, or 1000 μmol/L CaCl_2_). After that, cardiomyocytes were counted (repeating three times: 15 μL of cell suspension + 15 μL of trypan blue; 10 μL of the mixture was added into the hemocytometer chamber for living cardiomyocyte counting), calculated, and diluted in 1500 cells/100 μL or 3000 cells/100 μL with a cardiomyocyte plating medium (Opti-MEM + 2.5% FBS, 10 mM BDM, and 1% Pen/Strep). One hundred microliters of diluted cell solution was seeded into laminin (20 μg/mL)-precoated XF96 cell culture plates (V3 PS) or 96-well plates and cultured for 2 h at 37 °C, 5% CO_2_ for adherence. After that, the cells were used for treatments and a subsequent Seahorse XF Cell Mito Stress Assay.

### 4.3. TNFα Dose-Responsive Experiment of Mitochondrial Respiratory Function 

Plasma TNFα reaches a maximum value of 20 ng/mL in septic animals [88]. We have also shown that serum TNFα levels were ~11.5 and 1.2 ng/mL at 1 and 4 h after a lipopolysaccharide challenge in vivo, respectively [52]. Therefore, dose escalation studies of TNFα (0, 5, 10, and 20 ng/mL/min) were conducted on the mitochondrial bioenergetic response in cardiomyocytes isolated from male adult mice. Cardiomyocytes were plated in a laminin-precoated XF96 cell culture plate at 3000 cells/well and treated with different doses of TNFα for 1 h. The cells were then subjected to bioenergetic profiling. The experiment was repeated two times.

### 4.4. Mitochondrial Bioenergetic Response by Seahorse XF96 Cell Mito Stress Test 

A Seahorse Bioscience XF-96 instrument (Seahorse Biosciences, North Billerica, MA, USA) was used to measure OCR in the cardiomyocytes as we recently reported [87]. FCCP titration (0, 0.5, 0.75, 1, and 1.5 µM of FCCP) was performed on cardiomyocytes from male mice with cell densities of 1500 or 3000 cells per well. Cardiomyocytes (3000 cells/well) were treated with TNFα (10 ng/mL) or H_2_O_2_ (50 μM [45]) in the absence or presence of E2 (100 nM [45,55,56]) in a supplemented XF medium (5 mM glucose, 1 mM pyruvate, and 2 mM glutamine) for 1 h. The bioenergetic profile of cardiomyocytes was measured sequentially as baseline OCR, ATP-linked production by injection of 1 μM oligomycin (Oligo), maximal uncoupled respiration by adding FCCP, and non-mitochondrial respiration by injection of 0.5 μM rotenone (R) and antimycin A (A). Basal OCR was the last value before the oligomycin injection - non-mitochondrial OCR. Maximal OCR was the maximal measurement after using FCCP - non-mitochondrial OCR. Spare respiratory capacity was represented as the percentage of maximal OCR vs. basal OCR.

### 4.5. Measurement of Mitochondrial Membrane Potential 

Two hours after being cultured in a cardiomyocyte plating medium, isolated mouse cardiomyocytes were exposed to TNFα (10 ng/mL) +/− E2 (100 nM) for 1 h. The cells were then treated with a fluorescent probe JC-1 (1 μM, G-Biosciences, St. Louis, MO, USA) at 37 °C. JC-1 shows green fluorescence in cytosol as monomers and displays red fluorescence in mitochondria as dimers/aggregates. After 30 min of incubation, live-cell imaging on cardiomyocytes was taken using an Axio Observer Z1 motorized microscope (Zeiss, Oberchoken, Germany) with a 10× objective. Fluorescence intensity of red and green in individual cardiomyocytes was quantified using ImageJ (NIH). The ratio of red-to-green intensity represented the mitochondrial membrane potential.

### 4.6. Statistical Analysis 

All data (except Figure 1 and Figure 3A) were from independent experiments repeated at least three times in at least quadruplicate. The data were evaluated using an unpaired *t*-test or an analysis of variance (ANOVA) test if it passed the Shapiro–Wilk normality test, and were represented as the means ± SEM. For those data that did not follow a normal distribution (did not pass the normality test), the Mann–Whitney or Kruskal–Wallis tests were used and stated in each figure legend. These results are shown as box and whiskers plots with a dot for each individual measurement (the upper and lower borders of the box indicating the upper and lower quartiles; the middle horizontal line showing the median; and the upper and lower whiskers displaying the maximum and minimum values). The difference was considered statistically significant when *p* < 0.05. All statistical analyses were performed using GraphPad Prism (GraphPad, La Jolla, CA, USA).

## 5. Conclusions

In summary, we investigated mitochondrial bioenergetics in cardiomyocytes isolated from adult male and female mice upon TNFα or H_2_O_2_ exposure using the XF analysis. We found that cardiac mitochondria in female mouse hearts were more resistant to acute stress, with better respiratory function than that in male mice. E2 treatment protected cardiomyocyte mitochondria from male mice against acute inflammatory and oxidative stress. Our findings provide particularly important evidence to explain why females have better cardiac recovery than males after I/R, trauma, shock, or sepsis [25,29,42,43,89].

## Figures and Tables

**Figure 1 ijms-23-09312-f001:**
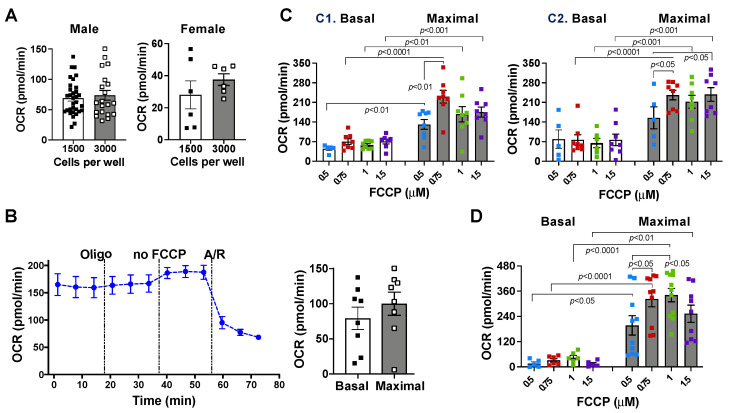
Effect of cell number and FCCP concentration on mitochondrial respiratory function in cardiomyocytes. (**A**) Basal oxygen consumption rate (OCR) in different cell densities. (**B**) The OCR trace and maximal OCR value without addition of FCCP in cardiomyocytes isolated from adult male mice. (**C**) Maximal OCR values with 0.5, 0.75, 1, and 1.5 M of FCCP in cardiomyocytes isolated from adult male mice at cell densities of 1500 (**C1**) and 3000 cells per well (**C2**), respectively. (**D**) Effect of FCCP concentration on maximal metabolic rate in cardiomyocytes isolated from adult female mice (3000 cells/well). Data were analyzed using the *t*-test or one-way ANOVA and represented as the mean ± SEM.

**Figure 2 ijms-23-09312-f002:**
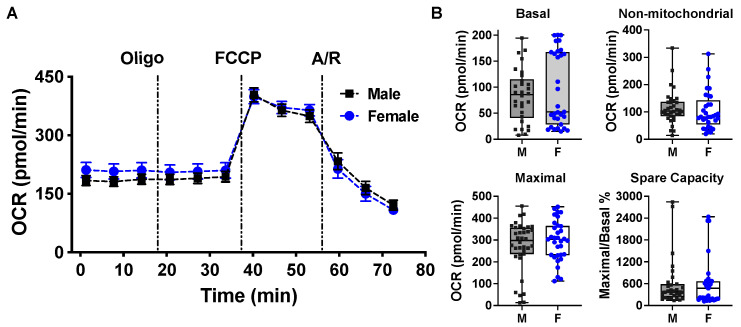
Metabolic profiling between cardiomyocytes isolated from adult male and female mice following addition of oligomycin (Oligo), FCCP, and Antimycin A/Rotenone (A/R). (**A**) The OCR trace. (**B**) Calculated values for mitochondrial basal, non-mitochondrial, maximal, and spare respiratory parameters between cardiomyocytes isolated from male (M) and female (F) mice. Data were evaluated using the Mann–Whitney test in B and shown as box and whiskers plots with a dot for each individual measurement. The results are from six mouse hearts of each sex in six trials.

**Figure 3 ijms-23-09312-f003:**
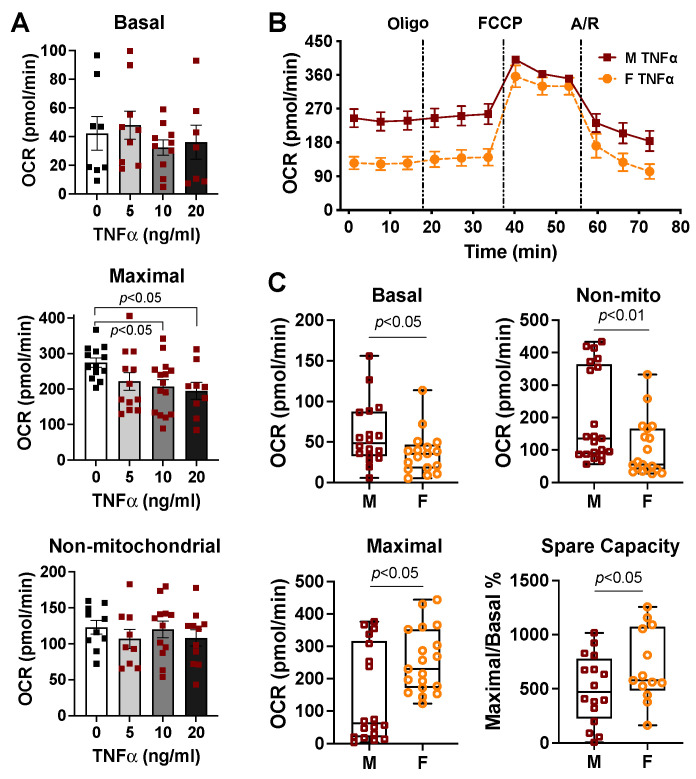
Sex differences in TNFα-changed metabolic profiling between cardiomyocytes isolated from adult male and female mice. (**A**) TNFα dose-responsive mitochondrial function: basal, non-mitochondrial, and maximal OCR in cardiomyocytes from male mice. Data were analyzed using one-way ANOVA with Dunnett’s multiple comparisons test and represented as the mean ± SEM. (**B**) The OCR trace in cardiomyocytes isolated from male (M) and female (F) mice following TNFα treatment (10 ng/mL, 1 h). (**C**) Quantification of basal OCR, non-mitochondrial OCR, maximal rate, and spare respiratory capacity in cardiomyocytes isolated from M and F mice subjected to TNFα (10 ng/mL, 1h). Data were evaluated using the Mann–Whitney test in C except for spare capacity (*t*-test) and shown as box and whiskers plots. The results from four mouse hearts of each sex in three trials.

**Figure 4 ijms-23-09312-f004:**
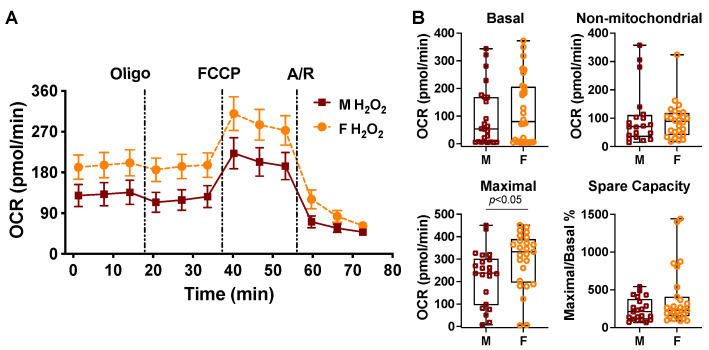
Sex differences in H_2_O_2_-altered bioenergetic response in cardiomyocytes isolated from male and female mice. (**A**) The OCR trace in cardiomyocytes from male (M) and female (F) mice following H_2_O_2_ (50 μM, 1 h) treatment. (**B**). Quantification of basal OCR, non-mitochondrial OCR, maximal rate, and spare respiratory capacity in cardiomyocytes from M and F mice subjected to H_2_O_2_. Data are analyzed using the Mann–Whitney test in B and represented as box and whiskers plots. The results were from four mouse hearts of each sex in four trials.

**Figure 5 ijms-23-09312-f005:**
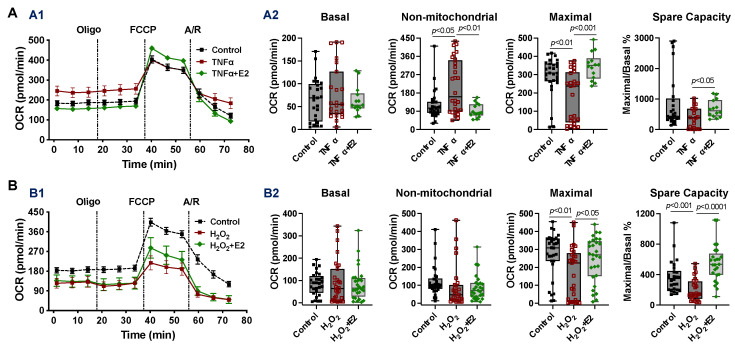
E2 restored mitochondrial respiratory function in cardiomyocytes isolated from male adult mice following acute stress of TNFα (**A**) or H_2_O_2_ (**B**) exposure. (**A1**,**B1**) The OCR trace. (**A2**,**B2**) E2 (100 nM) treatment restored TNFα- or H_2_O_2_-impaired mitochondrial respiration function (maximal OCR and spare respiratory capacity) in cardiomyocytes. Data were evaluated using the Kruskal–Wallis test in (**A2**,**B2**) and are shown as box and whiskers plots. The results were from at least three male mouse hearts in at least three trials.

**Figure 6 ijms-23-09312-f006:**
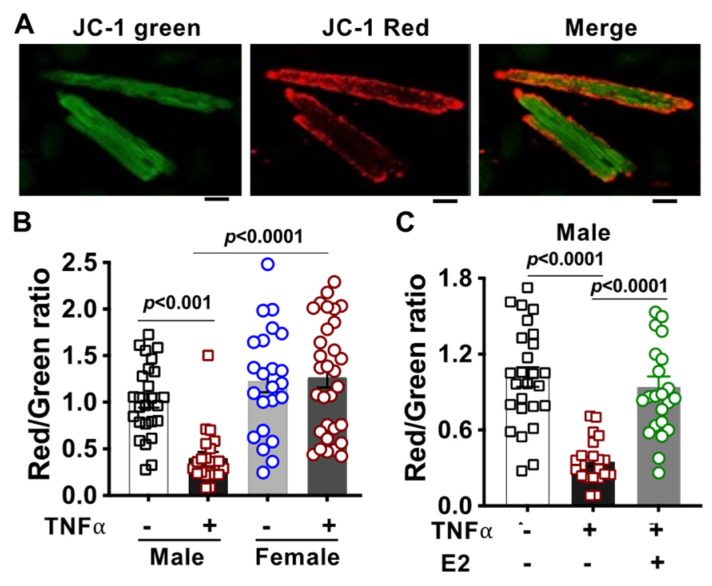
Effects of TNFα and E2 on mitochondrial membrane potential in adult cardiomyocytes. (**A**) Representative images of mitochondrial membrane potential using JC-1 in adult mouse cardiomyocytes. Scale bar = 20 μm. (**B**) Changes of mitochondrial membrane potential in cardiomyocytes isolated from male (M) and female (F) adult mice upon TNFα (10 ng/mL, 1 h) exposure. Two-way ANOVA analysis with a Tukey’s multiple comparisons post hoc test was used. The *p*-value was <0.0001 for factor of the sex, 0.0046 for factor of TNFα, and 0.0013 for the interaction between the two factors. (**C**). 17β-estradiol (E2, 100 nM) preserved TNFα-impaired mitochondrial membrane potential in cardiomyocytes isolated from male mice. One-way ANOVA with Dunnett’s multiple comparisons test was used and represented as the mean ± SEM. The results were from at least three mouse hearts of each sex in at least three trials.

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
