# Peer review of "Sex as Biological Variable in Cardiac Mitochondrial Bioenergetic Responses to Acute Stress"

_ijms, 2022, doi:10.3390/ijms23169312_

Round 1
Reviewer 1 Report
Through the manuscript entitled ‚Sex as Biological Variable in Cardiac Mitochondrial Bioenergetic Responses to Acute Stress‘ the authors Scott et al. aimed to investigate the impact of gender to the adult cardiomyocyte energy metabolism in acute stress condition. In general, to understand the differences in cardiac pathophysiology between men and women is of critical importance and merits further evaluation to develop targeted therapy not only between men and women, but to be able to define personalized therapy to a particular patient, concerning his/her metabolic state, actual diagnoses and onging therapeutical aprroaches. Here, the long-term interest of the group of Scott et al. is to understand the role of sex differences in the heart in various metabolic disturbances including ischemia/reperfusion injury, sepsis and trauma. Specifically for this case, based on their previous work well cited through the whole manuscript and focused on metabolic stress in the adult cardiac myocytes, they previously have found elevated levels of TNFalpha and in this work they analyzed the acute stress metabolic response to this importat cell stress marker as well as to well known stress inducer, the hydrogen peroxide. Moreover, they have found positive effect of the estradiol treatment, where they argue the estradiol may be the protective agent against stress in studied pathologies - ischemia/reperfusion injury, sepsis and trauma.
Although the manuscript is formally well written, as well as the reasoning to perform described experiments are in general important and/or based on previously obtained results, there are several critical issues which in my opinion make the results and conclusions elusive:
1. It is not stated, why the authors have selected the values of 1500 and 3000 cells per well. Why the OCR (Fig. 1A) was the same for 1500 cells as well as for 3000 cells condition? In the reference 47 the authors had much more higher myocyte density (6000-25000), moreover, from 6000 to 20000 myocytes per well the OCR increased linearly. Therefore there is strong expectation there should be clear difference between density of 1500 and 3000 myocytes per well in this study.
2. It is not stated neither in the manuscript nor in the supplementary material, how the myocyte density (1500 and 3000 myocytes per well) was estimated. This is crucial step to normalize the results for all data provided in the manuscript.
3. Statistical analysis – the authors state use of the SEM and t-test/ANOVA for all the data. However, in several cases, namely Fig. 2B (OCR Basal, OCR Maximal, Spare Capacity), Fig. 3C (OCR Non-mito, OCR Maximal), Fig. 5B (OCR Basal, OCR Maximal, OCR Spare Capacity), Fig. 5D (OCR Non-mitochondrial, OCR Maximal) is it hard to believe the data follow normal distribution. Information, how the normality of the data was tested is not provided. Re-analysis of the data regarding non-parametric statistics may change the results significantly. Independently, how the outliers, clearly visible in Fig. 2B (Spare Capacity), Fig. 5B (OCR Basal, OCR Spare Capacity) were considered? Moreover, in some cases it looks like there are appearing two populations of cells – see the Fig. 2B (OCR Basal, Maximal), Fig. 3C (OCR Non-mito, OCR Maximal), Fig. 5B (OCR Maximal), Fig. 5D (OCR Maximal), is there any explanation for this? In these cases the use SEM and t-test/ANOVA is confusing.
4. FCCP titration (Fig. 1C and Suppl. Fig. 1). Considering the data it looks like the concentrations of 0.75, 1 as well as 1.5 microM of FCCP induce very similar results.
5. Data provided in Fig. 6 are typical experimental design for two-way ANOVA approach. Therefore in Fig. 6B the effect of sex (M vs. F), effect of treatment (TNFalpha) and effect of interaction should be tested.
6. In fig. 6C, the female group treated with E2 should be implemented.
7. The authors observed no effect of Oligomycin, what implies the all oxygen consumption after Oligomycin application compensates the proton leak and is not used for ATP production. Is this then the experimental model of quiescent adult cardiomyocytes appropriate to study pathophysiological condition, as the authors declare ATP is critically important (lines 184, 190-191) to adapt to condition of TNFalpha/H2O2/E2 treatment?
Due to all the principal issues mentioned above, in my opinion the presented manuscript in its current form does not fill the criteria required for consideraiton to be published in the International Journal of Molecular Sciences. Attached are also some minor and formal concerns:
Minor issues
Line 76 – I suggest use term ‚rest‘ instead of ‚remainder‘
Line 101 - I suggest use term ‚ minimal/negligible‘ instead of ‚not much‘
Line 102 – the formulation ‚was also noticed‘ is confusing. Was there a significant difference in this case?
Fig. 5 – I recommend to merge the parts A and B in to A and then merge the parts C and D into B, as these parts of the figure desribe the same data
Fig. 5 – I recommend use the same symbols for control group – either ‚Control‘ or ‚M‘ for both Fig. 5B and Fig. 5D
Fig. 6 – I recommend to present representative images from other experimental groups as well
Formal concerns
Manuscript
Line 5-7 – adjust the state province in one style – either ‚IN‘ or ‚Indiana‘
Line 85 – the rest of the Fig. 2 Caption is missing
Line 138, line 262 – correct FCPP to FCCP
Line 152 – I suppose term ‚respiration‘ instead of ‚reparation‘ is correct in this context
Line 189 – remove comma in the text ‚ cardiomyocytes isolated from, adult male mice‘
Line 237 – correct ‚bioenergetics‘ to ‚bioenergetic‘
Supplementary material
Fig. 1 – the x axis has no marking – add ‚Time (s)‘ for both, A and B
Fig. 1, Caption, line 1 – correct FCPP to FCCP
Author Response
We sincerely thank you for reviewing our manuscript and for providing insightful suggestions. We have made all revisions along the lines suggested and addressed each below.
- It is not stated, why the authors have selected the values of 1500 and 3000 cells per well. Why the OCR (Fig. 1A) was the same for 1500 cells as well as for 3000 cells condition? In the reference 47 the authors had much more higher myocyte density (6000-25000), moreover, from 6000 to 20000 myocytes per well the OCR increased linearly. Therefore there is strong expectation there should be clear difference between density of 1500 and 3000 myocytes per well in this study.
Response: According to the literature, 6000 to 20000 cardiomyocytes per well seeded overnight on the 24-well Agilent Seahorse plate indicated lineal increase of the OCR values in the reference 49, whereas 1000 cells per well using the 24-well Agilent Seahorse plate were suitable to measure the OCR of cardiomyocytes within a couple of hours after their isolation in the reference 50. In this study, we performed XF Mito Stress test in several hours after cardiomyocyte isolation using 96-well Agilent Seahorse V3 plate. Therefore, we selected the cell density of 1500 and 3000 per well in our study. We have added this information in our revised manuscript (with track changes, lines 74 - 77).
The OCR values were comparable between 20000 and 25000 cardiomyocytes per well cultured overnight on the 24-well Agilent Seahorse plate (reference 49). Considering that much less (1000) cells per well were needed in short-time cultured cardiomyocytes for detection of mitochondrial bioenergetic response using the same XF24 Agilent Seahorse instrument (reference 50), it is possible that comparable basal OCR values between cell density of 1500 and 3000 myocytes per well using a XF96 Agilent Seahorse analyzer for short-time cultured cardiomyocytes in this study, were likely due to a decrease in sensitivity with many cells (reference 49).
Of note, based on the Agilent Seahorse XF recommendation, basal OCR ranges would be targeted between 20 and 165 pmol/min to avoid potential floor or ceiling values after using mitochondrial respiration modulators. Also, to minimize the effect of losing unattached cells on mitochondrial bioenergetic measurements, we selected higher cardiomyocyte density (3000 cells/well) that fell into the recommended basal OCR range in this study.
- It is not stated neither in the manuscript nor in the supplementary material, how the myocyte density (1500 and 3000 myocytes per well) was estimated. This is crucial step to normalize the results for all data provided in the manuscript.
Response: We are sorry for this omission. We have added this information as "cardiomyocytes were counted (repeating three times: 15 ml of cell suspension + 15 ml of trypan blue; 10 ml mixture was added into the hemocytometer chamber for living cardiomyocyte counting), calculated, and diluted in 1500 cells/100 ml or 3000 cells/100 ml with cardiomyocyte plating medium (Opti-MEM + 2.5% FBS, 10 mM BDM, and 1% Pen/Strep)" in the revised manuscript (with track changes, lines 293 - 297).
- Statistical analysis – the authors state use of the SEM and t-test/ANOVA for all the data. However, in several cases, namely Fig. 2B (OCR Basal, OCR Maximal, Spare Capacity), Fig. 3C (OCR Non-mito, OCR Maximal), Fig. 5B (OCR Basal, OCR Maximal, OCR Spare Capacity), Fig. 5D (OCR Non-mitochondrial, OCR Maximal) is it hard to believe the data follow normal distribution. Information, how the normality of the data was tested is not provided. Re-analysis of the data regarding non-parametric statistics may change the results significantly. Independently, how the outliers, clearly visible in Fig. 2B (Spare Capacity), Fig. 5B (OCR Basal, OCR Spare Capacity) were considered? Moreover, in some cases it looks like there are appearing two populations of cells – see the Fig. 2B (OCR Basal, Maximal), Fig. 3C (OCR Non-mito, OCR Maximal), Fig. 5B (OCR Maximal), Fig. 5D (OCR Maximal), is there any explanation for this? In these cases the use SEM and t-test/ANOVA is confusing.
Response: We apologize for this confusion. We did check data distribution using Shapiro-Wilk normality test in our original manuscript and used Mann Whitney test or Kruskal-Wallis test for those data not passed normality test. We have added this information to 4.6 Statistical Analysis (with track changes, lines 328 - 329) and figure legend. However, we did miss the normality test for Fig. 5A2-A5, 5B2-B5 and re-analyzed them using Kruskal-Wallis test.
We considered the outliers using ROUNT method at the beginning. Since we found that removing outliers did not change the statistical results, we kept these outliers in the figures and statistical analysis.
We agree with you that there are appearing two populations of cells in some figures. This could be attributable to batch effects. Cells responded better with higher values in some-day experiments, while the lower measurements were obtained in other day experiments. To minimize this batch effect, we isolated male and female cardiomyocytes at the same day and plated them in the same 96-well Agilent Seahorse plates for assay.
- FCCP titration (Fig. 1C and Suppl. Fig. 1). Considering the data it looks like the concentrations of 0.75, 1 as well as 1.5 microM of FCCP induce very similar results.
Response: We agree with you that similar results appear among 0.75, 1, and 1.5 mM of FCCP induced maximal OCR in male cardiomyocytes (Fig. 1C). However, there were no statistical differences in groups of 1 and 1.5 mM of FCCP vs. the 0.5 group (Fig. 1C1). Also, additional data from female cardiomyocytes (Fig. 1D) suggested that 0.75 mM of FCCP was a most suitable dose in this study. We have changed the wording to “0.75 µM of FCCP was suitable to induce maximal respiration response …” in this revised manuscript (with track changes, line 88).
- Data provided in Fig. 6 are typical experimental design for two-way ANOVA approach. Therefore in Fig. 6B the effect of sex (M vs. F), effect of treatment (TNFalpha) and effect of interaction should be tested.
Response: Thank you for bringing it to our attention. We have used two-way ANOVA to analyze data in Fig. 6B. Although statistical significance was observed between F C and M TNFa, we don't feel this difference could imply any important information and did not include it.
- In fig. 6C, the female group treated with E2 should be implemented.
Response: Thank you for the suggestion. We have performed additional experiment and include the data in supplemental figure 3.
- The authors observed no effect of Oligomycin, what implies the all oxygen consumption after Oligomycin application compensates the proton leak and is not used for ATP production. Is this then the experimental model of quiescent adult cardiomyocytes appropriate to study pathophysiological condition, as the authors declare ATP is critically important (lines 184, 190-191) to adapt to condition of TNFalpha/H2O2/E2 treatment?
Response: Based on the description of Seahorse XF Mito Stress Test for ATP production after using oligomycin, a potential decrease in OCR indicates the portion of basal respiration used for driving ATP production. In other words, no effect of oligomycin implies that mitochondrial oxygen consumption and substrate-consuming process during basal respiration are mainly mediated by proton leak in isolated cardiomyocytes. This information has been added to the revised manuscript (with track changes, lines 195 – 197). The Seahorse XF Cell Mito Stress Test is designed for assessing mitochondrial respiration function instead of ATP production, which can be measured using Agilent Seahorse XF Real-Time ATP Rate Assay kit (with track changes, lines 274 – 276). If mitochondrial membrane potential and normal proton gradient are altered within cellular context under pathophysiological condition, mitochondrial respiration function (essential for ATP production) could be changed. Therefore, we believe our experimental model of quiescent adult cardiomyocytes is appropriate to study mitochondrial respiration response during pathophysiological condition.
Due to all the principal issues mentioned above, in my opinion the presented manuscript in its current form does not fill the criteria required for consideration to be published in the International Journal of Molecular Sciences.
Response: Thank you again for your insightful suggestions. We believe that our revised manuscript has been significantly improved and met the criteria required for publication in the International Journal of Molecular Sciences.
Attached are also some minor and formal concerns:
Minor issues
Line 76 – I suggest use term ‚rest‘ instead of ‚remainder‘
Response: Thank you. We have changed it.
Line 101 - I suggest use term ‚ minimal/negligible‘ instead of ‚not much‘
Response: Thanks. We have used 'minimal/negligible'.
Line 102 – the formulation ‚was also noticed‘ is confusing. Was there a significant difference in this case?
Response: We have changed this sentence to ‘Notably, cardiomyocytes isolated from female mouse hearts displayed higher levels of maximal mitochondrial respiratory function in response to H2O2 compared to male cardiomyocytes.’ Yes, there was a significant difference in Fig. 4B Maximal.
Fig. 5 – I recommend to merge the parts A and B in to A and then merge the parts C and D into B, as these parts of the figure desribe the same data.
Response: Thanks. We have merged them as you suggested.
Fig. 5 – I recommend use the same symbols for control group – either ‚Control‘ or ‚M‘ for both Fig. 5B and Fig. 5D
Response: Sorry for this inconsistence. We have changed 'M' to 'Control' in Fig. 5A2.
Fig. 6 – I recommend to present representative images from other experimental groups as well
Response: We have provided representative images from each experimental group in supplemental figure 4.
Formal concerns
Manuscript
Line 5-7 – adjust the state province in one style – either ‚IN‘ or ‚Indiana‘
Line 85 – the rest of the Fig. 2 Caption is missing
Line 138, line 262 – correct FCPP to FCCP
Line 152 – I suppose term ‚respiration‘ instead of ‚reparation‘ is correct in this context
Line 189 – remove comma in the text ‚ cardiomyocytes isolated from, adult male mice‘
Line 237 – correct ‚bioenergetics‘ to ‚bioenergetic‘
Response: Thank you. We have changed them as you suggested.
Supplementary material
Fig. 1 – the x axis has no marking – add ‚Time (s)‘ for both, A and B
Fig. 1, Caption, line 1 – correct FCPP to FCCP
Response: Thank you. We have corrected them as you suggested.
Please also see our point-by-point response in the attachment.

Reviewer 2 Report
In the present manuscript, the authors showed the difference, in terms of sex, in mitochondrial bioenergetics following TNF and H2O2 exposure.
Overall it is a very interesting paper with a precise initial hypothesis, extremely adequate methods used, and a detailed description and demonstration of the results. Contratulations to the authors. I have no other comments.
Author Response
We sincerely thank you for reviewing our manuscript and for positive comments on our study/paper.
Reviewer 3 Report
This is quite an interesting study showing sex differences in cardiomyocyte response to stress induced by TNFa or H2O2 exposure. The authors show that female mice displayed better mitochondrial bioenergetic response to TNFa or H2O2 than those isolated from male mice. In this case, 17b-estradiol (E2) treatment protects male cardiomyocytes from induced stress. I have a few comments and questions:
1. On Fig. 1 authors need to add a legend showing FCCP concentrations. In the caption to the figure, the authors also need to correct FCPP to FCCP, the same applies to fig. S1.
2. Why did the authors conduct an experiment with a different number of cells? This should be explained immediately in the results section. Why it was performed only on male cardiomyocytes, but not females, although the article shows important sex differences in the response of the cells. The authors themselves note the need for FCCP titration of female cardiomyocytes, I think that this will not take much time and they should do these experiments. This will eliminate some of the contradictions indicated in the text.
3. I also have a methodological question. On Fig. 1A, the authors present an OCR curve whose data were used to calculate calculated values for mitochondrial basal, non-mitochondrial, maximal, and spare respiratory parameters (panel B). In this case, it can be seen that in panel B, the points have an extremely large scattering, which does not correspond to the OCR curve. What is the reason for this? In general, this question concerns all the presented figures.
4. Did the authors assess cell viability prior to the mito stress assay? This may have influenced the results of the study.
5. Did the authors evaluate the work of the respiratory chain complexes in the supercomplex mode? This can be discussed.
6. The column labels in fig. 6B and C should be more clear and obvious. Or it needs to be deciphered in the caption.
7. In addition, in Fig. 6 the authors need to provide representative microscopy data of both male and female cardiomyocytes.
Author Response
Please see our point-by-point response in the attachment.

Reviewer 4 Report
Can the authors comment on whether E2 treatment can be considered as an adjunct therapy for patients under acute stress?
Other than this, I have no additional comments or concerns for the study. The objective, and rationale of the study is clearly explained and presented, and the limitations of the study
Author Response
We sincerely thank you for reviewing our manuscript and for positive comments on our study/paper.
We have discussed the possibility of using E2 treatment as an adjunct therapy for patients under acute stress in our revised manuscript (with track changes, lines 242 – 248) as follows: “It is noteworthy that the rapid use of estrogen post injury provided therapeutic potential following trauma-hemorrhage [73-75], burn [76, 77], and sepsis [78, 79]. Our group also demonstrated the beneficial effect of post-ischemic E2 treatment on heart I/R [45, 80]. More importantly, one dose usage of E2 right after burn injury significantly improved cardiac function and preserved mitochondrial performance within 24 hours [77]. We further observed that E2 treatment post ischemia corrected male cardiomyocyte mitochondrial activity following I/R [45]. Together with our current findings, these studies strongly support to consider the rapid use of E2 as an adjunct therapy for patients under acute stress.”
Round 2
Reviewer 1 Report
In general, by implementing new experiments (Fig. S3, Fig. S5), providing relevant figures to the supplementary material (Fig. S1C, Fig. S4) and addressing relevant methodological and conceptual points in the text of the manuscript, the authors Scott et al. have improved the quality of the manuscript a lot. Moreover, in the cover letter the authors have appropriately answered most of the questions raised, especially regarding the lack of Oligomycin effect in the cardiac myocytes and relation to the ATP production (question #7) as well as the myocyte counting methodology (question #2) and choice of the 3000 myocytes per well (question #1). However, some of the questions merit further evaluation, namely:
Question 3, first part.
Response: However, we did miss the normality test for Fig. 5A2-A5, 5B2-B5 and re-analyzed them using Kruskal-Wallis test.
The authors confirmed the non-normal distribution of the data mentioned in the response. Then the presentation of the data using mean +/- SEM is not correct. The data following non-normal distribution have to be presented in form of median and quartiles (box and whiskers graphs). Please correct for all the data in the manuscript following non-normal distribution.
Question 3, third part: Moreover, in some cases it looks like there are appearing two populations of cells…
Response: We agree with you that there are appearing two populations of cells in some figures. This could be attributable to batch effects. Cells responded better with higher values in some-day experiments, while the lower measurements were obtained in other day experiments. To minimize this batch effect, we isolated male and female cardiomyocytes at the same day and plated them in the same 96-well Agilent Seahorse plates for assay.
Glad to read the authors have considered the issue with two populations of the cells. However, the quote ‘Cells responded better with higher values in some-day experiments, while the lower measurements were obtained in other day experiments. To minimize this batch effect, we isolated male and female cardiomyocytes at the same day’ is confusing. This implicates to me there is a hidden factor, which strongly influences the data obtained in a particular day. To avoid this, the authors proclaim to ‘isolate male and female cardiomyocytes at the same day’. Was then this approach helpful to minimize the effect the two populations? Are the data presented in the manuscript (Fig. 2B, Fig. 3C, Fig. 5B, Fig. 5D) after this improvement (i.e. isolating male and female myocytes at the same day)?
Question 5. Data provided in Fig. 6 are typical experimental design for two-way ANOVA approach. Therefore in Fig. 6B the effect of sex (M vs. F), effect of treatment (TNFalpha) and effect of interaction should be tested.
Response: Thank you for bringing it to our attention. We have used two-way ANOVA to analyze data in Fig. 6B. Although statistical significance was observed between F C and M TNFa, we don't feel this difference could imply any important information and did not include it.
Authors declare they have performed the two-way ANOVA statistical test for the data provided in the Fig. 6B. Authors reply ‘statistical significance was observed between F C and M TNFa’ is confusing – what was the p value of the sex, p value of the treatment (TNFalpha) and p value of the interaction between these two factors? These three p values have to be clearly stated in the manuscript. Moreover, when looking at the data provided, the authors most probably used the post-hoc tests between the groups Male TNFalpha(-) vs Male TNFalpha(+) as well as between Male TNFalpha(+) vs Female TNFalpha(+). Which post-hoc test they have used?
Regarding all the answers provided by the authors I consider the status of the manuscript of authors Scott et al. as Minor changes. Some formal issues pending:
Fig. 6A and Fig. S4 – calibration bars are missing. In Fig. S4 there is stated the magnification was 150x, however, such description does not follow the publication standards for nowadays image information data. Calibration bar with indication of the length of the calibration bar has to be provided. In principle it would be good to provide this information also in the Fig. S1C, however, in this case the images are just demonstrative, therefore calibration bars in this case are not critically required.
Line 77 – specify in more detail the term ‚several hours‘, i.e. ‚from four to six hours‘ or similar.
Line 129 – the comment usage of the term ‘minimal/negligible‘ was meant either to use term ‚minimal‘ or term ‚negligible‘.
Line 295 – correct ‚caculated‘ to ‚calculated‘
Line 297 – correct ‚humdred‘ to ‚hundred‘
Add SC to the Abbreviations list
Author Response
We sincerely thank you for reviewing our manuscript again and for acknowledging the improvement of our revised manuscript. We have made the revisions based on your helpful suggestions and addressed each below.
--- Question 3, first part.
The authors confirmed the non-normal distribution of the data mentioned in the response. Then the presentation of the data using mean +/- SEM is not correct. The data following non-normal distribution have to be presented in form of median and quartiles (box and whiskers graphs). Please correct for all the data in the manuscript following non-normal distribution.
Response: Thank you for pointing it out. We have changed the way to present the data with non-normal distribution from mean +/- SEM to the form of median and quartiles (box and whiskers graphs) in the revised manuscript (Fig. 2B, 3C, 4B, 5A2 and 5B2), and have added this information to the figure legend and Statistical Analysis section (lines 315 – 317).
--- Question 3, third part: Moreover, in some cases it looks like there are appearing two populations of cells…
Glad to read the authors have considered the issue with two populations of the cells. However, the quote ‘Cells responded better with higher values in some-day experiments, while the lower measurements were obtained in other day experiments. To minimize this batch effect, we isolated male and female cardiomyocytes at the same day’ is confusing. This implicates to me there is a hidden factor, which strongly influences the data obtained in a particular day. To avoid this, the authors proclaim to ‘isolate male and female cardiomyocytes at the same day’. Was then this approach helpful to minimize the effect the two populations? Are the data presented in the manuscript (Fig. 2B, Fig. 3C, Fig. 5B, Fig. 5D) after this improvement (i.e. isolating male and female myocytes at the same day)?
Response: Isolation of adult cardiomyocytes is a tricky process. Cellular status of adult cardiomyocytes is influenced by many factors during isolation, including enzyme activity difference (from different lot number), heart flow rate under the Langendorff of constant pressure, and small embolism in the heart vessels, etc. This approach could minimize the batch effect between male and female cardiomyocytes from different day experiments. Yes, the data presented in the manuscript (Fig. 2B, Fig. 3C, Fig. 5A2, Fig. 5B2) are after this improvement.
--- Question 5. Data provided in Fig. 6 are typical experimental design for two-way ANOVA approach. Therefore in Fig. 6B the effect of sex (M vs. F), effect of treatment (TNFalpha) and effect of interaction should be tested.
Authors declare they have performed the two-way ANOVA statistical test for the data provided in the Fig. 6B. Authors reply ‘statistical significance was observed between F C and M TNFa’ is confusing – what was the p value of the sex, p value of the treatment (TNFalpha) and p value of the interaction between these two factors? These three p values have to be clearly stated in the manuscript. Moreover, when looking at the data provided, the authors most probably used the post-hoc tests between the groups Male TNFalpha(-) vs Male TNFalpha(+) as well as between Male TNFalpha(+) vs Female TNFalpha(+). Which post-hoc test they have used?
Response: The p value is <0.0001 for factor of the sex, 0.0046 for factor of TNFa treatment, and 0.0013 for the interaction between the two factors. We have added this information to the Fig. 6 legend. Yes, we used Tukey’s multiple comparisons test between groups, which has also been added to the revised manuscript.
--- Fig. 6A and Fig. S4 – calibration bars are missing. In Fig. S4 there is stated the magnification was 150x, however, such description does not follow the publication standards for nowadays image information data. Calibration bar with indication of the length of the calibration bar has to be provided. In principle it would be good to provide this information also in the Fig. S1C, however, in this case the images are just demonstrative, therefore calibration bars in this case are not critically required.
Response: We have added scale bars in Fig. 6A and Fig. S4 to follow the publication standards for nowadays image data and provided this information in the figure legend. Since Fig. S1C is just demonstrative, calibration bars are not provided.
Line 77 – specify in more detail the term ‚several hours‘, i.e. ‚from four to six hours‘ or similar.
Response: Thank you. We have provided this information (line 74).
Line 129 – the comment usage of the term ‘minimal/negligible‘ was meant either to use term ‚minimal‘ or term ‚negligible‘.
Response: We are sorry about the misunderstanding and select ‘negligible’ in the revised manuscript.
Line 295 – correct ‚caculated‘ to ‚calculated‘
Line 297 – correct ‚humdred‘ to ‚hundred‘
Response: We have corrected them.
Add SC to the Abbreviations list
Response: We have added it.
Reviewer 3 Report
The authors have significantly improved the presentation of the data. I have no more questions and comments.
Author Response
We sincerely thank you for reviewing our manuscript again and for recognizing the improvement of our revised manuscript.